# A Categorical Model of General Consciousness

**DOI:** 10.3390/biomimetics10040241

**Published:** 2025-04-14

**Authors:** Yinsheng Zhang

**Affiliations:** Institute of Scientific and Technical Information of China, Beijing 100038, China; meta-math@outlook.com

**Keywords:** model of consciousness, homomorphism of consciousness, general consciousness, conscious Turing machine, recursion in neuron networks, cognizance

## Abstract

Consciousness is liable to not be defined in scientific research, because it is an object of study in philosophy too, which actually hinders the integration of research on a large scale. The present study attempts to define consciousness with mathematical approaches by including the common meaning of consciousness across multiple disciplines. By extracting the essential characteristics of consciousness—transitivity—a categorical model of consciousness is established. This model is used to obtain three layers of categories, namely objects, materials as reflex units, and consciousness per se in homomorphism. The model forms a framework that functional neurons or AI (biomimetic) parts can be treated as variables, functions or local solutions of the model. Consequently, consciousness is quantified algebraically, which helps determining and evaluating consciousness with views that integrate nature and artifacts. Current consciousness theories and computation theories are analyzed to support the model.

## 1. Introduction

Since the new millennium, biomimetics gradually has been undergoing a paradigm shift to simulate consciousness, which is evidenced by more and more technologies emerging or under precast, germane to consciousness. For example, Jangsun Hwang, etc., expect the future technologies of biomimetics, including aircraft, automobiles, and robots; [1] “Getty Images”, to involve brain-reading robots [2]. These technologies are not intended for simulating parts of biological organs as in traditional biomimetics, but to simulate, or physically duplicate, recognize, and understand the will of a whole body. Besides the expected biomimetics technologies that closely relate to consciousness, current biomimetics technologies are seeking to characterize consciousness to a large extent. For example, Kyle Mizokami states that “The drone is an example of biomimetics, in which machinery, particularly drones, are designed to resemble living creatures in order to take advantage of the animal’s physical advantages” [3]. Accordingly, “Biomimetics” has been re-defined as follows:

Biomimetics is an interdisciplinary field in which principles from engineering, chemistry and biology are applied to the synthesis of materials, synthetic systems or machines that have functions that mimic biological processes.[4]

According to this definition, simulating consciousness such as AI, at least the will-simulating technologies should be classified into an interdisciplinary field, for consciousness serves as a natural, biological process. Therefore, we might accept that consciousness-simulating biomimetic technologies such as drones are emerging.

Consciousness is a complex object of philosophy, science, and common sense. However, science sometimes leaves the defining of consciousness to philosophy and considers the object only relative to or as the biological basis of [5] consciousness, regardless of remarkable advances of consciousness. Meanwhile, biomimetics and AI seem to essentially emulate consciousness, which calls for not (only) a philosophic but also a scientific definition of consciousness, so there should be a clear and tangible definition of consciousness, as in what in-physical and everyday consciousness refers to. In as much as consciousness is empirical and appeals to science, both philosophy (conservatively, as a part) and common sense do not refute consciousness as a scientific object. That is, in contrast to meta-physics, science should be “philosophy-inclusive” and at least provide an answer to what the relationship of meanings of consciousness between philosophy and science is. In fact, the paradigm of studying consciousness is evolving such that science is not only addressing the question of “what is the basis of consciousness?” [5] but also of “what is consciousness?”. As science studies nature, consciousness as a natural activity is spontaneously included as no extra-nature exists, and it should not be treated only as a basis to the concept mentioned by philosophy, as a supervenience to philosophy, as epiphenomena, or as only explananda of the concept. Hence, it is indispensable to scientifically define consciousness in detail, mathematically, physically, biologically, or even technically for biomimetics, AI, etc., which should be compatible overall with the deep-rooted concept regardless of the disciplines in which it is used.

The relationship of consciousness between philosophy and science was proposed by Uriah Kriegel as follows:

Philosophy may have a more significant role to play in shaping our understanding of consciousness; that even a complete science of consciousness may involve certain lacunae calling for philosophical supplementation.[6] (pp. 1–13)

This paradigm presents an architectural scheme that naturally (physically, chemically, biologically, etc.) analyzes phenomena corresponding to philosophical ideas. It establishes a scheme where philosophy explains consciousness based on the hypothesis that the philosophical ideas on consciousness are relatively clear, not controversial for an integrated and objective world. However, as philosophy is too imaginary and always controversial, the paradigm calls for an improvement, namely the acceptance of natural analysis in philosophy. This paradigm is, as proposed here, a mathematics assistant; that is, a mathematical framework where its infusing methods are comprehensive, parallel to natural analysis and also to philosophy, is desired. The present study attempts to establish an algebraic model that serves as such a framework and as methods to mathematically and, therefore, epistemologically describe and analyze consciousness, including the relevant philosophical ideas. Against its controversial definitions or interpretations, mathematical methods may achieve the “Greatest Common Divisor”—the widest consensus drawn from various statements on consciousness by modeling common and intrinsic features inside the statements and descriptions with greater precision.

D. M. Armstrong and Norman Malcolm [7] (pp. 3–45) abstract two semantics of consciousness, which summarize the essence of consciousness well. One is the transitivity, meaning sensing or being aware of objects; another one is the non-transitive, meaning a mental state yet not unconscious. If we regard the non-transitive as the capability of performing the transitive, we can simplify consciousness as the transitivity of experiential objects or some mental states. Therefore, could we acquire a mathematical description of consciousness in the sense of transitivity? Fortunately, in this way, mathematically modeling consciousness has been attempted in the past. Piaget, the psychologist influenced by the mathematical school Bourbaki, established a mathematical model to depict consciousness generated from the growth of a young child. Piaget’s consciousness model properly reflects the essence of consciousness—transitivity with established psychological logic [8,9,10].

Piaget gave his definition of consciousness in terms of “cognizance”, a type of behavior that is a psychological structure holding a logic where a cognized object is homomorphic to the type:

In general, when a psychologist speaks of a subject being conscious of a situation, he means that the subject is fully aware of it.

Cognizance (or the act of becoming conscious) of an action scheme transforms it into a concept and therefore that cognizance consists basically in a conceptualization.

Thus, cognizance, starting from the periphery (goals or results) moves in the direction of the central regions of the action in order to reach its internal mechanism: recognition of the means employed, reasons for their selection or their modification en route, and the like.[8] (pp. 332–352)

Piaget explained the mechanism of cognizance as that depicted in Figure 1 [8] (pp. 332–352), where the interaction (indicated by the arrows) between the subject and object is given, accompanying double centers as the cores of the corresponding classifications and the abstracted environment.

Concerning homomorphism between the subject and object with their own operations or movements, Piaget gave an example of setting up a causality of consciousness [11]:

Returning to the problems with which we begin, one can wonder what the relationships are between the correspondences or transformations that the child discovers in reality and those that he discoverers in his own operations or actions. In particular, one can ask whether there exists, as we suppose, a correspondence between causality contributed to objects and the subject’s operatory structure.

Correspondences of this sort between causality and the subject’s deductive productions become conscious only tardily, of course.

Identical to Malcolm referring the essence of consciousness to the transitivity (“being aware of”) between a psychological state and the corresponding objects, Piaget’s cognizance interaction between the two sides of the periphery is transitivity; moreover, the pairs in interaction, Center and Center’, maintain their own transitivity on a level, and the double transitivity on their own levels constructs a homomorphism. Thus, we can express Piaget’s categorical idea about consciousness [12] as (1), with the corresponding diagram shown in Figure 2.Csc = ({Rft, Csc}, *F*: Rft → Csc)(1)
where Csc denotes consciousness, the object of Rft denotes referents, and the morphism is *l*(*A*,*B*), *m*(*C*,*D*), *A*,*B* ∈ Csc, *C*,*D* ∈ Rft (*l* represents a psychological operation like “logic” as Piaget described, and *m* represents the objective “movement” or behavior of the subject).

The objects and morphisms satisfy the following: *f*(*A*,*B*) × *f*(*B*,*C*) → *f(A*,*C*) (transitivity—Piaget typically instantiates the transitivity as operations which have identical or equivalent effects to the composition) and Iden (*A*,*A*) (identity).

*F* is a natural transform, which resembles sensing objects (like a projection of, say, a ball) and generates the corresponding mind, which are mappings of the objects onto psychological states. *F* represents the transitivity mentioned by Malcolm. The morphisms *l* and *m* and the transformer *F* are all compositing/ed or expanding/ed such that the image is flexible in functional selves, in the levels, and for the levels.

Equation (1) exhibits a homomorphism between referents and their subjects, or consciousness and its reflected objects in a categorical scheme, as stated in (2).Rft ≅ Scs(2)

In other words, although (1) and (2) do not describe an inner characteristic of what is to be determined as consciousness, like features or structures, they give a composite relation structure by which consciousness can be determined by these outer relationships (like *l* and *F* in Figure 2). Thus, it is the relationship, varying with each other between the information *F*(*A*) as the reflex in the line of logic *l* and the periphery worlds (an object *A*, referred to by a mind event *F*(*A*)), confirmed by a function or effect, that consciousness is defined in languages and thoughts, including philosophic or scientific. In particular, Piaget described consciousness as realizing some logic operations like inferences by the process; a reflex (like *F*(*A*)) in the brain resulted from an input gives rise to, say, predicting a position of the referent object (say, *C* in Figure 2 after a movement of *A*). Thus, a simple stimulus–reaction can be mapped by *F,* but it lacks sufficient relevant operation logic *l* to predict a movement of stimulus (*m*), causing such an excessively simple reaction to not always be determined as a conscious event. Similarly, unconsciousness might go through *F* but lack sufficient {*l*}, causing *F*(*A*) to fail in becoming a conscious event made up of {*F*(*B*), *F*(*C*)}. The transitivity on one layer like *F*(*A*) → *F*(*C*) guarantees equivalency arising in the operations on the layers, so logic can be generated----it is the generation of logic that is beyond simple *F* that might be not linking to {*l*}. 

Equations (1) and (2) show a paradigm of the mathematical modeling of consciousness; in parallel, there exists a paradigm of Kriegel [6], where mathematics is weaker than Piaget’s model, and philosophy—its notations and questions—inspires science (including that of consciousness) to be confirmed and resolved. The two schemes of Piaget and Kriegel, as visually represented by (a) and (b) in Figure 3, respectively, call for an evolved paradigm so that mathematic and philosophic approaches are all sufficiently applied in defining and explaining consciousness, as proposed in Figure 3c.

## 2. A Three-Layer Categorical Model of General Consciousness (3-CMGC)

### 2.1. The Outline of 3-CMGC

Equation (1) with Figure 1 omits materials (neurons, machine units, etc.) processing the mappings from outer objects into psychological states for generating consciousness, which need to insert a material layer to represent the substance implementing the processing.

Equation (3) with Figure 4 attempts to insert such a material layer, Unt, representing material bodies, like neurons and artifact units.

The novel Equation (3) serves as an improvement in homomorphism as expressed by (4).Rft ≅ Unt ≅ Csc(3)

Equation (3) defines consciousness as a kind of psychological or artificial state grounded in phenomena of living matter, like neurons, or software; Equation (4) shows a homomorphism between consciousness Csc and its reductive matters Unt, and its referents (objects projecting onto Rft, like the periphery of the subject).

Equations (3) and (4) are the result of the epistemological paradigm of modeling consciousness as stated in (**c**) in Figure 3, which belongs to a monism that regards consciousness as the identity of both psychologic and material entities, and the two entities are representable by virtue of physical methods.

When the model applies to artificial objects like robots, AI, and artificial consciousness, the artifacts are viewed as conscious. Thus, the model is viewed as one of general consciousness, called a three-layer categorical model of general consciousness (3-CMGC).Csc = ({Rft, Unt, Csc}, *F*_1_, *F*_2_: Rft → Unt → Csc)(4)

### 2.2. A Direct Depiction of Reductive Physicalism

Equations (3) and (4) are also a result of the epistemological paradigm of modeling consciousness, and they belong to a reductive physicalism that regards consciousness as the phenomenon of the material entity—for unambiguity from the term “materialism” that excessively highlights social meanings, “physicalism” is chosen while holding the essentially material characteristics of consciousness with the principle of materialism departing from the social meanings.

The reductive physicalism of consciousness, typically as elaborated by Francis Crick [13] and Zenon W. Pylyshyn [14] in an attempt to fill the gap in explanation between mental or similar properties and physical properties or between the mind and brain, represented here between Csc and Unt, has been refuted, where at least not all consciousness can be essentially subsumed under physical (biological, chemical, etc.) phenomena; that is, consciousness is physically inexplicable, or partly is, forever [15] (pp. 455–484) [16] (p. 117).

This refutation, however, faces objection such as from David Papineau [17] (pp. 14–36). Here, we further verify that the functor Csc ↔ Unt ↔ Rft, linked with consciousness Csc with an exact correspondence of neurons Unt, and optionally linked with the objective periphery world Rft, exist truly, wherein Csc ↔ Unt primarily implies the identification of the two factors deemed by monism and reductive physicalism.

Verifying the existence of Unt for biological bodies involves determining the neurons {*F*_1_}, which definitely play a role in deputizing certain objects to participate in brain thinking to complete some functions of generating consciousness.

The idea that a physical (biological) unit represents an object has existed for a long time and is called “Psychophysics”, advocated for by Friedrich Albert Lange. Neo-Kantianism regards the sensation from objects as “The transcendent basis of our organization remains therefore just as unknown to us as the things which act upon it. We have always before us only the product of both” [18], which implies that organs especially represent objectives, or certain objectives may own their own cells in the organs in terms of the schemes Csc ↔ Unt.

The subsequent development of science has confirmed this prediction that existing neurons play a role in standing for special kinds of objects. In particular, some space-representative neurons are explored.

In 1959, David Hubel and Torsten Wiesel revealed that in a cat’s various striate cortex areas (receptive fields), discharging varies with changes in angles of the stimuli, so certain cells are space–feature-related [19].

In this case, let ***α*** indicate the angles that are stimuli in David Hubel and Torsten Wiesel’s experiment, so *F*_1_(***α***) indicates that the receptive fields of single neurons in the cat’s striate cortex definitely react to ***α***. Suppose that the cat had a conscious activity *F*_2_. Thus, a function between Rft, Unt, and Csc can be expressed as (5), with reference to Figure 4.




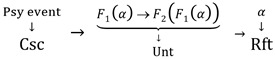

(5)



Another discovery of space-positioning cells in the brain of a rat was achieved by John O’Keefe, as well as May-Britt Moser and Edvard Moser in 1971 and 2005, respectively. Let *α*_1_, *α*_2_, and *α*_3_ denote the direction, position, and velocity of a space object as stimuli, respectively, in their experiment; then, (5) still holds with the Rft variables *α*_1_, *α*_2_, and *α*_3_ in place of *α* [20,21,22,23].

Recent similar discoveries determining neurons of orientation and positioning by means of functional magnetic resonance imaging (fMRI) have been made [24,25,26].

A visual object projected onto the neurons holding and acting as its primary topological geological characteristics was unveiled later by Tootell, Roger BH et al., which makes a composition of neurons with a 2D-topological shape identical to the shape of the object. This displays well a fractional (omitting the transitivity like {*l*}, {*u*} in a layer) homomorphism between Rft and Unt [27,28].

The neurons representing an object not only in neuron reflection but also in thinking by means of using the representing neurons are called “Mirror Neurons”, which was discovered by M. Fabbri-Destro and G. Rizzolatti [29]. This indicates an intact homomorphism between Rft and Unt.

As time and space are not only objective parameters in physical laws but also subjective consciousness functions, as Kant systematically came up with his theory that reason is consciousness reorganizing sensory information by time and space [30] (p. 61), it is believed that the existence of neurons definitely acts as certain objects and plays a representing role in thinking to form consciousness.

### 2.3. A Reductive Physicalism Explanation of Qualia

As stated in Section 2.2, the reductive physicalism of consciousness in an attempt to fill in the claimed gap in explanation between mental properties and physical properties is being doubted, wherein an important point is that the sensation, or qualia, of a process, upon a pain-resulting stimulus, as a salient example, is taken as unexplained, as some insist that nociceptive-specific neurons firing is regarded as pain being unintelligible [17]. The present study proposes that this gap is overstated, and it is physically explainable as it is semantically calibrated. It is noteworthy that “consciousness” is a notation in use in the context of thoughts, and its explanation is limited in use in languages so that the language makes sense. Thus, consciousness is a referrer that is different from the referent such as objects like the cosmos or an apple because a referrer provides an explanation for the use in context, and a referent meets objective laws (The relationship between the referrer and the referent is delineated in (Scs, Unt) and Rft in Figure 4). Then, the physical explanation, particularly upon the troubling problem qualia, should suffice the semantics of “consciousness” in terms of referring. This means that the first semantics of transitivity in [7] (pp. 3–45) is more essential in the definition of consciousness. Hence, we should take a position that the second semantics of consciousness, without referring to something, should be treated as a confirmation of the transitive usage of “consciousness” rather than a consciousness itself. That is, one in pain is conscious of referring (secondly aware of the pain), and when one cannot say “being conscious of a molecular of a certain chunk of neurons in pain”, it means that the result of the pain is conscious rather than the pain per se. Thus, the result is physically explainable with physical means, and a result, the subject’s conscious experience of semantically explaining consciousness consists of the relationship in the three-layer construct. Therefore, a narrowed gap is explainable with and for physical features.

## 3. The Construct of Consciousness in View of 3-CMGC

3-CMGC raises the question of whether it can develop, or has contained, a construct of consciousness.

Constructs and compositions of consciousness have been uncovered by many philosophers and psychologists, most notably Kant, who regarded the mind as a cognition process which conforms to objectivity with physical laws, the paradigm of the Cognitive Theory of Consciousness (CTC). In addition, he referred to his thoughts as a “future scientific metaphysics”, which includes the contemporary cognition science cross-disciplines of philosophy, psychology, AI, neuroscience, etc., as remarked by Matt Maccormick [31], where it had been argued by Kitcher, Brook, Sellars, and others that Kant’s philosophy of mind makes valuable contributions to contemporary cognitive science and artificial intelligence.

Kant viewed consciousness as a complex process which follows receptivity and understanding to generate knowledge and its synthesis. He thought of consciousness as only matter as thoughts, or a form of knowledge resulting from appearance [32]. These thoughts interpret the process of consciousness in line with the description in Figure 5.

For explaining meanings of (represented as modules in Figure 5) functions in mind, Kant gave definitions or statements, among which some key notations are as follows:

Concerning Module 1 [30] (pp. 59–61):

Whatever the process and the means may be by which knowledge refers to its objects, intuition is that through which it refers to them immediately, and at which all thought aims as a means. But intuition takes place only insofar as the object is given to us.

The capability (receptivity) to obtain representation through the way in which we are affected by objects is called sensibility. Objects are therefore given as by means of our sensibility. Sensibility alone supplies us with intuitions.

The effect produced by an object upon the capacity for representation, insofar as we are affected by the object, is sensation.

I call all representation pure (in a transcendental sense) in which there is nothing that belongs to sensation. this pure form of sensibility may it self be called pure intuition. These belong to pure intuition, which even without an actual object of the senses, exists a priori in the mind, as a mere form of sensibility.

There are two forms of sensible intuition, which are principles of a priori knowledge, namely space and time.

Concerning Module 2:

We call sensibility the receptivity of our mind to receive representation insofar as it is in some wise affected, while the understanding, on the other hand, is our faculty of producing representations by ourselves, or the spontaneity of knowledge.[30] (p. 86)

The thought of consciousness appealing to reason, as analyzed in Section 2.3, indicates that sensation, which contains qualia, not wholly serves as consciousness but as an intermediary nature extracted for generating consciousness. However, in the former process of receptivity and understanding, there are infused priori functions, which are priori intuition and priori logic; these two should be classified into consciousness. This is because this priori intuition, referred to by Kant as time and space to tidy up intuition, has been confirmed as participating in the generation of consciousness by means of a time-intermediary “script” [33] or as the cognition of the space feature or remaining as that introduced by Section 2.2. In addition, that logic, which a human holds, is confirmed as consciousness by Piaget [11]. Therefore, the developed Kant’s model of mind should be conceivable in (3) and (4), such that (3) and (4) obtain a more detailed categorical form as in (6).({*F*_1,2_,_…*n*_}, →, (→, →)) ≅ (***F***, →, (→, →))(6)
where → denotes the meaning of logic implication; (→, →) denotes morphisms like *A*→*B*, *B*→*C*, *A*→*C*, or a linear process of objects by natural laws; ***F*** are natural laws, which support the functions in *F*_1,2,…,*n*_; and {*F*_1,2,…,*n*_: *x*→*y*} are the individual psychological contents, basically in long-term memories as the factors in Global Workspace Theory (GWT [34]), like logic, knowledge bases, or culture in terms of Kant’s priori generating or appealing to reason and science, such that a conscious event *F*_i_ calls {*F*_1,2…,*n*_}, as (7) shows*F_i_* ∈{*F*_1,2…,*n*_}(7)
where {*F_i_*} are specific experiences, as originated from the object with their features in Figure 5, which accompany their individual long-term psychological contents like logic, knowledge, or culture as the priori function.

Consequently, Kant’s model of consciousness can be represented as a homomorphism as Figure 6, which is a simplified version of Figure 4 for simplifying a transitivity in a layer.

After converging (Neo-)Kant’s model with the correction from Piaget of turning the priori into generation in constructing consciousness, 3-CMGC starts to delineate a consciousness even with a static framework of a consciousness construct as stated in (6) and (7), which should be an addition to the models of consciousness based on the Cognitive Theory of Consciousness (CTC) and GWT to be an inner construct to add (3) and (4).

## 4. The Computability of Consciousness in View of 3-CMGC

### 4.1. Current Computability Models of Consciousness

Now that consciousness is regarded as a cognitive activity according to the CTC, it should be computable, as explained by Charles Wallis, where the exact nature of cognition lies in dynamic computationalism [35].

Consciousness has actually been viewed as computable for a long time since the proposition by the Turing machine (TM). Strong AI at least favors this proposition, as mentioned by David J. Chamers:

The strong AI thesis is cast in term computation, telling us that implementation of the appropriate computation suffices for consciousness. To evaluate this claim, we need to know just what it is for a physical system to implement a computation.[36]

In addition, he predicted that a computation implementing consciousness would be a TM in the form of combinational-state automata (CSA).

Recently, consciousness-represented brain neurons have been modeled by Lenore Blum and Manuel Blumin in more detail as a conscious Turing machine (CTM) [37]. This model proposes that conscious neurons according to GWT are represented by neuron networks, and the connexity of the networks is regression, that is, Turing-computable. The CTM describes consciousness-related neurons as a six-tuple:<STM, LTM, Up Tree, Down Tree, Links, Input, Output>(8)
where

STM: Short-term memory that, at any moment in time, contains the CTM’s conscious content;

LTM: Long-term memory, like expertise constituting processors;

Env → LTM: Given that edges are directed from the environment via sensors to processors of the sensory data, Env is an environment for the neurons;

LTM → STM: Via the Up Tree;

STM → LTM: Via the Down Tree;

LTM → LTM: Bidirectional edges (links) between processors;

LTM → Env: Edges directed from specific processors (like those that generate instructions for finger movement) to the environment via actuators (like the fingers that receive instructions from these processors) that act on the environment (via the actions of the fingers from these processors).

According to the theory of the CTM, the stage is represented by STM that, at any moment in time, contains the CTM’s conscious content. The audience members are represented by an enormous collection of powerful processors—each with its own expertise—that make up LTM processors that make predictions and receive feedback from the CTM’s world. Based on this feedback, learning algorithms internal to each processor improve that processor’s behavior. Thus, LTM processors, each with their own specialty, compete to obtain their questions, answers, and information in the form of chunks in the stage for immediate broadcast to the audience.

Conscious awareness is defined formally in the CTM as the reception by the LTM processors of the broadcast of conscious content. In time, some of these processors become connected by links turning conscious communication (through STM) into unconscious communication (through links) between LTM processors. Communication via links about a broadcasted chunk reinforces its conscious awareness, a process known as ignition.

A conscious neuron phenomenon is explained as a chunk of linked nodes in the Up Tree—an up-directed binary tree—as the chunk wins the competence of neurons through a function **f** that holds, expands, or curtails its nodes such that the chunk acquires sufficient |weight| and weight, respectively, called the intensity (the sum of the sub-node weights after a duration) and mood—a node’s weight by virtue of assigning probability in a coin-flip neuron node. Therefore, f is regressive and, thus, Turing-computable.

CTM neuron dynamics of a conscious event show a more competitive neuron network in a GWT. Spontaneously, it raises two points as follows:iIf a (pure) mathematical representation of a CTM can be made, note that the mode of the TM is an extractive mechanism model rather than a (pure) mathematical one (compare notations like (of transition function) “turning left”, “turning right”, “tape writer”, and “tape reader” in the TM with “prime number” and “triangle” in pure mathematics);iiIf there exists a common model depicting consciousness of both the brain and machines (if the machine or artificial consciousness is putative), note that the mode of the CTM explains only biological brains, or a similar mechanism model rather than a mathematical model.

### 4.2. A Proposed Computability Model of General Consciousness of 3-CMGC

In this section, a computability model of general consciousness is proposed to respond to points (i) and (ii).

The Church–Turing thesis was proved as a theorem by Yuri Gurevich [38] with the help of Marvin Lee Minsky [39], which proves that a TM equivalently serves as a general recursion function [40,41,42], defined by Godël [43], making up primitive recursions in a recursion or a composition pathway as expressed by (9) and (10), wherein the general recursion function *f* is achieved by means of recursion of *h* and *g*, and *x* is a vector.*f*(***x***, 0) = *g*(***x***)(9)*f*(***x***, *y* + 1 = *h*(***x***,*y*, *f*(***x***,*y*))(10)

Equations (9) and (10) describe a definite decision and are different from rough or flexible values such as {*l*}{*u*} in an interval stage as logic in generation described by Piaget, as a script for a final version of a conscious event described by Deniel Dannett, or as a moving scheme in a competence for winning as the CTM predicts. Moreover, a swarm computation is not targeted by Equations (9) and (10), which forms the reason for the emergence of AI’s new strands—connectionism.

Upon this insufficiency in depicting rough or flexible values in a network computation, a mathematical model of neuron network computation is proposed here. Consider a kind of machine learning, an artificial neural network (ANN), which simulates natural neurons. Common formulas of this ANN can be expressed as (11) and (12).{*f*_0_(***x***,0)} = {*g*(***x***)}(11){*f*_2_(***x***,***y*** + 1)} = {*h*(***x***, ***y***, *ε*, *f*_1_(***x***, ***y***))}(12)
where ***x*** are certain elements in Rft; *g* initializes the informalization of ***x*** for input as nodes of neurons in the 0 layer; {*g*(***x***)} indicates inputting examples of *g*(***x***) many times in training; *f*: ***x*** → *f*(***x***) is a recursion function, for example, the reflex function of nodes; *f*_0_(***x***, 0) obtains the connection weight of elements of ***x*** from the 0 layer of the ANN; {*f*_0_(***x***, 0)} indicates inputting *f*_0_(***x***, 0) many times in training; *f*_1_ = ***w*^T^ *x*** + *ε* + ***y*** obtains the connection weight of elements of ***x*** with bias *ε* based on ***y***; ***y*** is the regression vector of neurons next to ***x***; *f*_2_ is the recursion function for the next layer of *f*_1_; {*f*_2_} refers to *f*_2_ for multiple training; *ε* is a bias for adjusting the weights of nodes ***w***; and *h* is a threshold function, including iterating *ε* many times in training {*h*}.

Equations (11) and (12) give the result of a neuron network {***x***, ***y***} in recursion satisfying (9) and (10), constituting a general recursion of multiple training. Considering the neuron network given by (8), it is plausible that the CTM can be expressed as (11) and (12).

It is determined from (11) and (12) that a learned outcome {*f*} on the primary structure {*g*} in a similarity emerges, as indicated by (13).{*f*} ≈ *L*({*g*})(13)
where {*f*} learns, viz. *L* from {*g*}, which is a proposed relation beyond the Godël recursion model (Equations (9) and (10)).

The anatomical evidence of the human (brain) neuron network (HNN) holding an identical structure to the ANN and showing the same mechanism in learning has been given and modeled by Kali and Dayan [44].

More detailed evidence that the HNN is identical in construction to the ANN was recently confirmed by Jane X. Wang et al., who claimed that there exists a learning system—a meta-reinforcement learning system in the prefrontal cortex of the human brain [45]—where even the corresponding living matter of the weights in the HNN is narrowed upon the participation of dopamine (DA), which is the molecular-level evidence mapping the molecular Unt onto the Csc, which can be formalized in (14) and (15).{Consc_0_(photons,0)} = {info(photons)}(14){Consc_2_((photons), *y* + 1)} = {*h*(Consc(photons, *y*, amount(DA), Consc_1_(photons, *y*))}(15)

Equations (14) and (15) are instances of learning of the HNN-ANN, which contain a core of recursion in swarm computation (second-order variables of sets in contrast to (11) and (12)). Corresponding to 3-CGCM, (14) and (15) implement the following diagram from a learning unit to the classified result:


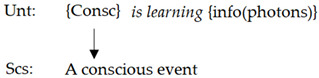
or a diagram with three layers as in (16).




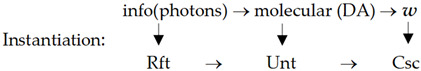

(16)



Equations (13)–(16) describe sets of recursion variables in place of individual variables as in (9) and (10) and for brains or machine learning of a general consciousness of species of natural and artificial units. Hence, points i and ii should be considered solved. This implies that the swarm characteristic variables have been joined as a TM model, such that cognitive theories based on swarm characteristic variables in terms of the TM are mathematically represented. For example, “multiple scripts” [34] for editing into consciousness can be mathematically represented as the type {*f*} in (11)–(13).

### 4.3. The Biological Positive of Recursion for 3-CMGC

Empirical arguments that confirm that neurons carry out natural computation to perform a recursion were authenticated. Here, we cite a study by H. R. Heekeren and S. Marrett et al., who conducted an experiment that made subjects decide whether stimuli were a face or a house given an image with noise [46].

With the decrease in noise in images showing a face or a building, the subjects’ brain fMRI signals in an area of their dorsolateral prefrontal cortex (DLPFC) are proportional to the gaps of cognition cells corresponding to a face or house, which shows the decision of neurons to subtract to maximize a characteristic difference. If “intensity” means the intensity of fMRI signals for a special object (face or building), using “areas” to denote a reflecting region in the subject’s brain, the experiment demonstrates a homomorphism as in (17).




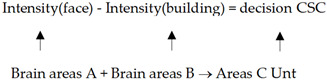

(17)



The quantification relation in (17) (up formula) is a mathematical expression, which is a primitive recursion function implemented by the brain neurons (dictated by down expression). This shows an instance of the material layer Unt (neurons and their properties) running computations {*u*} and being homomorphic to Csc (like making a decision).

Recently, increasingly more discoveries of the functional neurons have been achieved, where mathematical features are verified to be gradient-characteristic and -relative for functional neutrons [47], which implies natural computations belonging to {*u*}. The relations between the two gradients are primitive recursion (projection), as (18) shows.


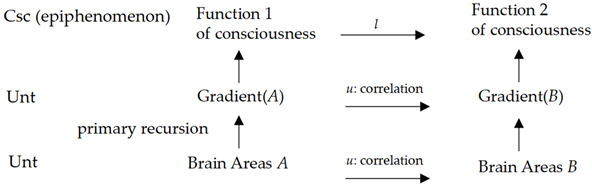
(18)
where the recursion function is of regression, that is, running in two steps. If an input ***α*** in Rft is assumed, the mappings between layers are given as (5), showing that the Unt operation is a composite.

## 5. The Unity of Natural and Artificial Consciousness in 3-CMGC

Functionalism and computationalism and, therefore, physicalism in view of the present study, all argue that the mind, particularly consciousness, is computable, according to which there exists a common algorithm like (11) and (12) depicting a machine or artificial system functionally equivalent to a natural consciousness. Recent advancements show an attempt to establish an artificial consciousness model with the extracted pillars—data, information, knowledge, wisdom, and purposes (DIKWP) [48,49]—which has a similar essence to physicalism in the unity of artificial and natural consciousness, much like 3-CMGC with the idea of homomorphism. The viewpoint of unity seems to be reasonable; however, there is still a “conservative” instance that insists that the gap between natural consciousness and technologies claiming “possessing consciousness” cannot be filled, as held by John Search [16] and Stephen Rothman [50], for example, the characteristic of the first person with privacy is treated as one of the essential factors of consciousness, which does not concede physicalism like models (3) and (4). Here, we give an argument that “the first person” is accessed, not private.

Consider the case of a brain–computer interface (BCI). An experiment was created to read the mental contents from brain activity through fMRI [51]. In the experiment, brain fMRI data were recorded while the subject viewed a large collection of natural images *I*. The data were used to estimate a quantitative receptive-field model *F*_2_ for each voxel for space, orientation, and spatial frequency from responses evoked by stimuli of the natural images. The model *F*_3_ predicts brain activity that sees potential images *I’* that match the stimuli images *I* with a maximum correct ratio of 92%. In other words, the images in Rft were being coded in Unt by brain fMRI as *F*_1_ to generate image variables *I* in Csc by the brain-receptive-field models, say, *F*_2_; *F*_2_ was to be estimated by the brain-receptive-field models *F*_3_, giving the potential (objective) images *I*’ in Csc’ (machine consciousness). This result confirms that *I* ≈ *I*’ with a maximum correct ratio of 92% as if Csc ≈ Csc’, as shown in Figure 7.

As a result, private Csc becomes open to a large extent (92%) as Csc-simulated Csc’. In other words, the experiment accessed the brain of the subject by means of fMRI, and its signals were verified as consciousness about space. Then, the first-person characteristic of consciousness was broken, becoming open to a large extent.

Similarly, collecting signals from the brain, regardless of interference, semi-interference, or non-interference in a BCI, may all be regarded as breaking the “first person” for breaching the private and making the brain’s function common. In other words, a BCI fully authorizes first-person access, and their outputs, mostly with appliable outcomes, show a conscious result such as an everyday action, which undermines the monopoly of the “first person” of biological entities, which delivers pursuable evidence to confirm the unity of general consciousness by filling the gap between artificial and natural consciousness.

## 6. Conclusions

Mathematically modeling consciousness is feasible. Although consciousness (as the specific mind) is considered the most complex in the world, and many aspects of consciousness are hidden, it is reasonable to set up a framework determining the essence of consciousness. According to transitivity, we propose here the essential meaning based on common sense and learning from Piaget’s two-layer consciousness model, 3-CMGC, of homomorphism between the objects and the materials implementing consciousness and the consciousness per se, which is not only for living matter, say, neurons, but also for artifacts.

The expectation of 3-CMGC presents a space classifying local and stepwise studies of consciousness into a global view. That is to say, a specific organ, areas of a brain, chunks of neurons, or computational parts in a machine, similar to or claimed as consciousness, can be functionally determined by a role making up 3-CMGC, and the determination can be quantified as if the function of the objects in question satisfies the algebra of 3-CMGC. In other words, 3-CMGC is a frame in which functions can be determined as a local solution as a kind of conscious phenomena, no matter of a machine or a biological body, by any experimental or quantified approaches. For example, Figure 7 shows a recognition of images by reading and simulating consciousness by a machine, with quantification methods to model a function in homomorphism between real images in recognition of neurons and generated images by computer algorithms. Besides determining consciousness by specifying the relationship of functions on and mapping across the layers in 3-CMGC, inner structures carrying out functions on a layer in 3-CMGC are characterized by Kant’s mind model in Figure 5 and by the improved computation model of biological and artificial neuron networks as shown by (14) and (15). Therefore, it is believable that 3-CMGC should be beneficial for applications in quantitative determinations of what measure biological and machine matters obtain consciousness, contributing to the paradigm shift of biomimetics.

## Figures and Tables

**Figure 1 biomimetics-10-00241-f001:**
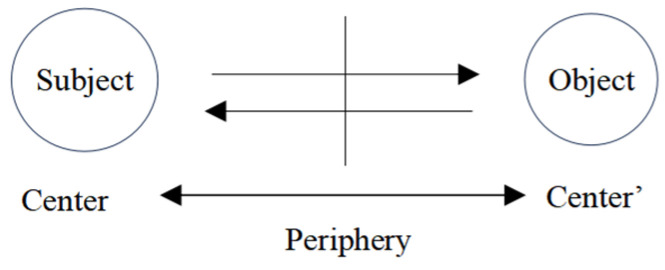
The cognizance mechanism set by Piaget.

**Figure 2 biomimetics-10-00241-f002:**
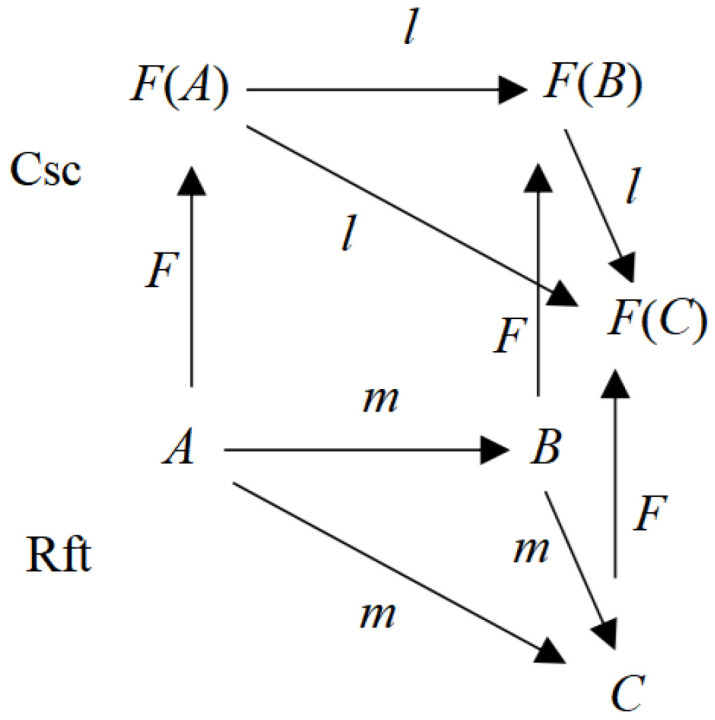
A categorical model of consciousness extracted from Malcolm and Piaget.

**Figure 3 biomimetics-10-00241-f003:**
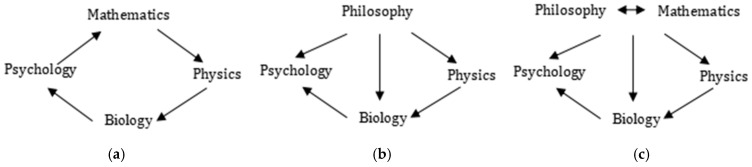
An epistemological evolution of modeling consciousness from (**a**) Piaget and (**b**) Kriegel to (**c**) a new complex paradigm proposed by the present paper.

**Figure 4 biomimetics-10-00241-f004:**
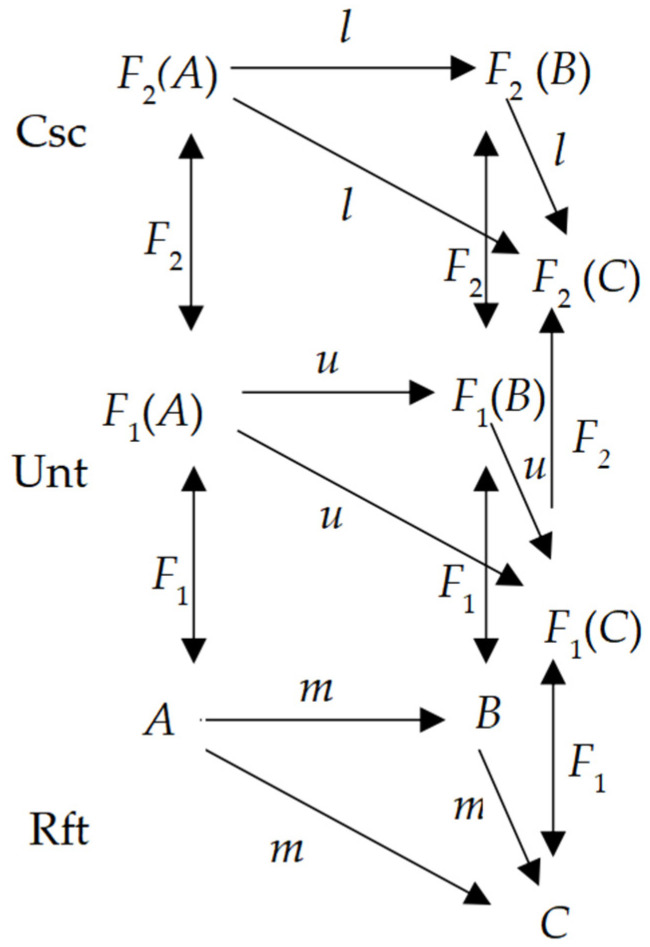
A 3-layer categorical model of general consciousness (3-CMGC) with a novel layer of substance Unt.

**Figure 5 biomimetics-10-00241-f005:**
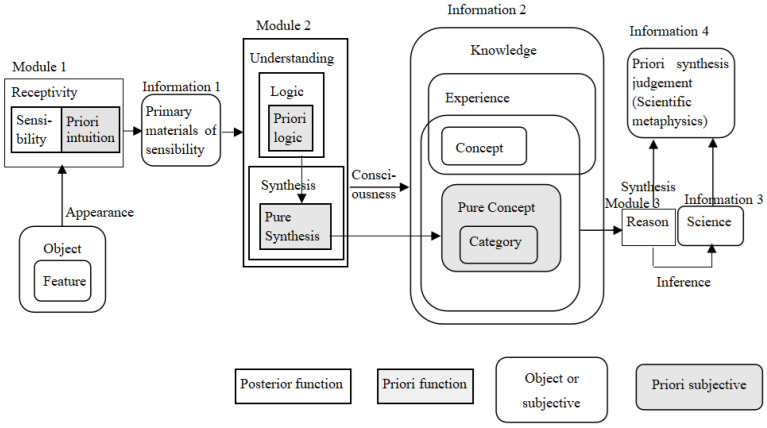
Diagram of Kant’s mind model with the consciousness process.

**Figure 6 biomimetics-10-00241-f006:**
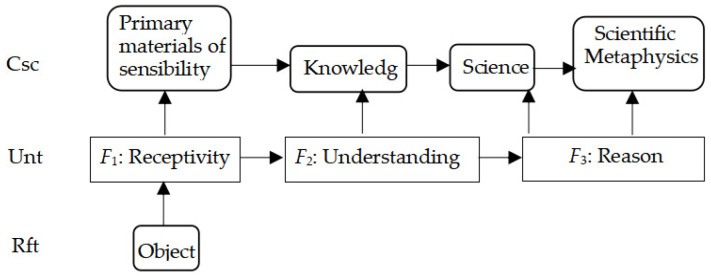
A diagram of Kant’s mind model with the consciousness process.

**Figure 7 biomimetics-10-00241-f007:**
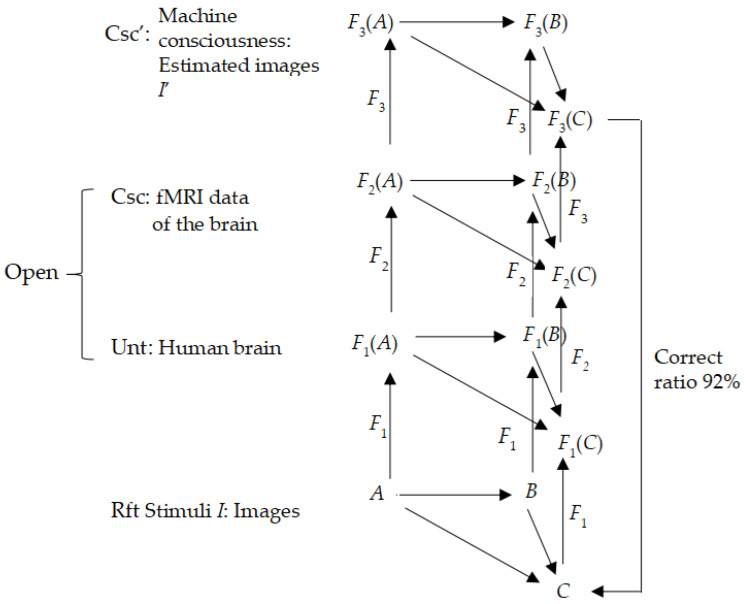
Reading images in thought with a correct ratio of 92% confirms that the first person is broken open as a third person for the machine reader.

## Data Availability

The original contributions presented in the study are included in the article, further inquiries can be directed to the corresponding authors.

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
