# Peer review of "A Categorical Model of General Consciousness"

_biomimetics, 2025, doi:10.3390/biomimetics10040241_

Round 1

Reviewer 1 Report

Comments and Suggestions for Authors

1. why use two different citation style? one [ ] and other is superscript of [ ].

2. "In this section, a computability model of general consciousness is proposed to respond to points (i) and (ii)." there is no (i) nor (ii) but only bullet points above?

3. not sure about the scientific proof your proposed model without more numerical evaluation or comparison against the previous known ones. 

Author Response

Dear reviewer:
    Thank you for your reviewing!
    After your review, I have made an improvement, which mainly remedies a argument that consciousness modelling is beneficial for a paradigm shift of biomimetics to simulate consciousness. The improvement consist in the following three points:
    (1) Three paragraphs are added at the begin, which introduces a context of biomimetics upon consciousness research.  
    (2) In the conclusion, and inner the contents of the paper, the relationship between biomimetics and the consciousness research is introduced too. 
    (3) The figures are updated.

    Concerning the proposals or questions of the reviewer, I hereby give the answers:

"1. why use two different citation style? one [ ] and other is superscript of [ ]."
Answer: When a citation appears in a sentence, namely, as a word in this sentence, then this citation bracket is normal, like [1]; if a citation is not be mentioned in a sentence, that is , not as a word in the sentence, then the bracket serves as a superscript.

"2. "In this section, a computability model of general consciousness is proposed to respond to points (i) and (ii)." there is no (i) nor (ii) but only bullet points above?"
Answer: the two points are in the last section (section 4.1). And I revise the mention as "''...to points (i) and (ii) mentioned at the end of Section 4.1."

"3. not sure about the scientific proof your proposed model without more numerical evaluation or comparison against the previous known ones. "
Answer: my proposed model (3-CMGC, as Figure 4) is based on the previous models of consciousness such as Figure 2 extracted from Malcolm and Piaget, and Figure 3 from Piaget and Kriegel. The model of 3-CMGC is a frame, namely flexible for the various functions (numerical evaluations) of a special consciousness. The proof applies the current advances of consciousness studies, as given in Figue 7 (to meet Figure 4).

    The improved manuscript file is named as "A Categorical Model of General Consciousness V5", as attached here.
    If you have any question, please let me know, and I shall do my best to make improvements to meet the requirement of the special issue.

Best
Yinsheng Zhang

Reviewer 2 Report

Comments and Suggestions for Authors

What are the criteria for choosing all three layers of objects, material reflex units, and consciousness itself in homomorphism

The boundary conditions are missing  for The construct of consciousness given 3-CMGC

List all the evaluation parameters

Write the mathematical equations to support the model

Validation of the model in the study is needed

Sources of images and data for fMRI data of the brain and

Refer to good work and use in reference as you are using MRI

Study and analysis of different segmentation methods for brain tumor MRI application.

Revise some part of conclusion with proper quantified outcome

Author Response

 Response is attached as the file "Answers to reviewer 2". The revised version hereby attached is version 5 of "A Categorical Model of General Consciousness".

Reviewer 3 Report

Comments and Suggestions for Authors

Dear Yinsheng Zhang,

thank you for your extremely interesting manuscript. The idea to mathematically model consciousness is very exiting. 

I have two recommendations for you.

First, it might be good to check sentence construction. Some sentences are quite hard to understand and understanding would benefit from a clearer or simpler sentence structure. I marked one example in the manuscript (first page).

Second, it would be helpful to use author names of the articles you cite. Sometimes you refer to literature by the author names and sometimes by the reference number. Uniformity would be good.

Furthermore, the figures on page 13 are incomplete. Somehow not all parts of the figures are shown.

Best regards

Comments on the Quality of English Language

The english language is good, but sentence structure is to complex and hinders understanding.

Author Response

Dear Reviewer:

Thank you for your recommendations!

Concerning your two points, I hereby give the answers as following.

  • The sentences are checked again, and I made some revisions. As the paper of version 4 are edited by the Author Services of the journal (I notice that you might reviewed the version 1?), thus, would this version (version 5) be acceptable?
  • The citation forms are revised, which are by names of authors of the references.

If you have any requirement or question, please let me informed, I shall do my best to meet your requirements!

Best regards

Yinsheng Zhang

Round 2

Reviewer 1 Report

Comments and Suggestions for Authors

thanks for the updates

Author Response

Please see the answer to reviewer 1.

Reviewer 2 Report

Comments and Suggestions for Authors

The authors have not highlighted the revisions or are not clear from which line the work is done.

The validation part in point 5 is not clear and the suggestion is not included in the manuscript properly.

Not able to track all revisions properly.

Author Response

Please see the answer to reviewer 2 attached here.
